# Effectiveness Evaluation of a Primary School-Based Intervention against Heatwaves in China

**DOI:** 10.3390/ijerph19052532

**Published:** 2022-02-22

**Authors:** Yonghong Li, Bo Sun, Changlin Yang, Xianghua Zhuang, Liancheng Huang, Qingqing Wang, Peng Bi, Yan Wang, Xiaoyuan Yao, Yibin Cheng

**Affiliations:** 1China CDC Key Laboratory of Environment and Population Health, National Institute of Environmental Health, Chinese Center for Disease Control and Prevention, Beijing 100021, China; liyonghong@nieh.chinacdc.cn (Y.L.); sunbo@nieh.chinacdc.cn (B.S.); wangyan@nieh.chinacdc.cn (Y.W.); yaoxy@chinacdc.cn (X.Y.); 2Dongtai Center for Disease Control and Prevention, Dongtai 224299, China; ycl000118@163.com (C.Y.); zhuang891007@163.com (X.Z.); 3Yancheng Center for Disease Control and Prevention, Yancheng 224002, China; huang170466733@163.com; 4Jiangsu Provincial Center for Disease Control and Prevention, Nanjing 210009, China; wqq-djy@163.com; 5School of Public Health, The University of Adelaide, Adelaide, SA 5005, Australia; peng.bi@adelaide.edu.au

**Keywords:** difference-in-difference analysis, extreme heat, health education, intervention, primary school student

## Abstract

Background: Evidence of the effectiveness of intervention against extreme heat remains unclear, especially among children, one of the vulnerable populations. This study aimed to evaluate the effectiveness of a primary school-based intervention program against heatwave and climate change in China to provide evidence for development of policies for adaptation to climate change. Methods: Two primary schools in Dongtai City, Jiangsu Province, China, were randomly selected as intervention and control schools (CTR registration number: ChiCTR2200056005). Health education was conducted at the intervention school to raise students’ awareness and capability to respond to extreme heat during May to September in 2017. Knowledge, attitude, and practice (KAP) of students and their parents at both schools were investigated by questionnaire surveys before and after intervention. The changes in KAP scores after intervention were evaluated using multivariable difference-in-difference (DID) analysis, controlling for age, sex, etc. Results: The scores of knowledge, attitude, and practice of students and their parents increased by 19.9% (95%CI: 16.3%, 23.6%) and 22.5% (95%CI: 17.8%, 27.1%); 9.60% (95%CI: 5.35%, 13.9%) and 7.22% (95%CI: 0.96%, 13.5%); and 9.94% (95%CI: 8.26%, 18.3%) and 5.22% (95%CI: 0.73%, 9.71%), respectively, after intervention. The KAP score changes of boys were slightly higher than those of girls. Older students had higher score changes than younger students. For parents, the higher the education level, the greater the score change, and change in scores was greater in females than in males. All the health education activities in the program were significantly correlated with the changes in KAP scores of primary school students after intervention, especially those curricula with interesting activities and experiential learning approaches. Conclusions: Heat and health education program in primary school was an effective approach to improve cognition and behavior for both students and their parents to better adapt to heatwaves and climate change. The successful experience can be generalized to respond to the increasing extreme weather/climate events in the context of climate change, such as heatwaves, and other emergent occasions or public health education, such as the control and prevention of COVID-19.

## 1. Background

Global climate change, with increased frequency, intensity, and duration of extreme heat events, has and will continue to lead to more morbidity and mortality of heat-related diseases and injuries [1]. It was reported that the number of attributable deaths due to climate change was above 150,000 in the year 2000, of which 88% were children younger than 5 years old in both developed and developing countries [2]. Children are a particularly vulnerable population to climate change due to immature physiological and cognitive functions, poor acquired functional immune response, higher exposure per unit body weight, and spending more time outdoors [3,4]. The impacts of climate change on children’s health include direct and indirect adverse effects of extreme weather events, increased heat stress, reduced air quality, altered disease patterns of climate-related infections, and water, food, and nutrition insecurity in vulnerable areas [5,6].

Extreme heat has become one of the main causes of deaths in many regions [7]. Many epidemiological studies have reported the adverse health impacts of hot temperature and heatwaves [8]. Moore et al. [9] reported that children in the United States had a higher risk of heat-related death than the general population, whereas another study [5] found that infant heat-related mortality rate in the US may increase by 5.5% for girls and 7.8% for boys by the end of the 21st century under a business-as usual scenario. Several studies have found an increase in pediatric hospital admissions and emergency department (ED) visits during extreme heat events [10,11,12], and that about one-third of the annual exertional heat illness cases in EDs in the US from 2001 to 2009 were teenage male athletes [11]. Our previous multi-city study on heat and ED visits in China found that young people under 18 years had the highest risk from a 1 °C increase in the daily maximum temperature in summer [12].

Heat-related illnesses are preventable to some extent. Prompt and effective implementation of adaptation actions will protect human health from the effects of heat waves. Intervention strategies that focus on addressing threats of heat waves have been implemented in many countries, especially in North America, Europe, Australia, and parts of Asia [13,14]. The prominent strategies are heat health warning systems with risk communication and health education implemented in communities, targeting vulnerable populations and their carers [13,15,16]. However, there has been limited evidence about their evaluations [15,17], especially among children. A London study [15], for example, targeted the elderly, whereas an evaluation of the Heat Wave Intervention Program in China also focused on people over 15 years old, and the improvement in their knowledge, attitude, and practice [16]. Although intervention plans aimed at community residents’ prevention may also typically benefit children, no specific study has targeted children to assess the effectiveness of heat wave implementation programs.

To provide scientific evidence for local health and education authorities for adaptation to extreme heat among school children, we implemented a primary school-based intervention program to improve knowledge, attitude, and practice (KAP) related to extreme heat among school children, and evaluated the effectiveness of the intervention in Dongtai City, Jiangsu Province, China. We hope the findings can have policy implications for other countries and regions with similar climatic situations.

## 2. Methods

### 2.1. Study Area

We performed a primary school-based intervention study in Dongtai City, which is located in the central coastal area of Jiangsu Province in China, at 120°07′~120°53′ east longitude and 32°33′~32°57′ north latitude, with a total area of 3175.67 square kilometers and a permanent resident population of proximately 980,000. It belongs to the subtropical monsoon maritime climate, and has four distinct seasons with cold winters and hot summers. The annual average temperature is 15.0 °C in Dongtai.

### 2.2. Study Design

Two primary schools located in two communities with similar economic and education levels were selected in Dongtai on the basis of their willingness to participate. One school was randomly chosen as the intervention school and the other one as the control school. Considering their understanding ability, all the students from the third to the fifth grades at each school were chosen as the study population. In addition, the parents of these students were also invited to participate in this study.

A series of health education activities were carried out at the intervention school to raise the primary school students’ awareness of and ability to respond to climate change and heatwaves from May to September in 2017 (Appendix A). The main health education activities included running related health education courses, launching lectures and class meetings on climate change adaptation and children’s health protection, and holding competitions involving blackboard newspapers, drawing, and writing. The school clinic was required to prepare enough medicine to prevent and treat heatstroke in summer. Enough warm boiling water was provided to ensure that students could drink clean warm water at all times. Students were supervised to help develop the habit of washing hands before eating and after using the toilet. During the school summer holiday, the students were required to participate in the practice activities against climate change or heatwave at least three times, such as observing and writing down how their families or friends respond to the heatwaves. The students at the intervention school were also required to spread the knowledge on climate change and health impacts to their families. Furthermore, early warning information on heatwaves and health risks was also sent to the parents through mobile phone text messages to protect the health of students during the summer holiday. The percentages of students participating in various health education activities at the intervention school are shown in Appendix A. There were no intervention activities at the control school.

### 2.3. Questionnaire Survey and Data Collection

A structured questionnaire was used to assess the knowledge, attitude, and practice of both students and their parents in the adaptation to extreme heat and climate change. The pre- and post-intervention questionnaire surveys were carried out in May and October 2017, respectively. Each questionnaire survey was performed simultaneously at the intervention and control schools, for both primary school students and their parents. Each participant signed a written informed consent before each survey.

The answers to each question in the questionnaire were scored quantitatively (see Appendix A). The full score of knowledge (K), attitude (A), and practice (P) was 8, 3, and 4, respectively. The total KAP score was the sum of the scores of these three components, with the full score of 15.

Ethical approval for this study was granted by the National Institute of Environmental Health, Chinese Center for Disease Control and Prevention (Approval number: 201610). This project has been registered in Chinese Clinical Trial Registry (Registration number: ChiCTR2200056005).

### 2.4. Data Analysis

Difference-in-difference (DID) analysis is a popular method to evaluate the effectiveness of interventions [18,19,20]. In this study, DID analysis was conducted to examine the differences in KAP scores of the students and the parents after intervention between the intervention school and the control school. We established the dummy variable (time) for before and after intervention and the dummy variable (group) that distinguished the intervention school and control school. The time was considered 1 (time = 1) after intervention and 0 (time = 0) before intervention. The intervention school was assigned as 1 (group = 1), and the control school as 0 (group = 0). The DID variable was set by multiplying group and time. In addition, the individual factors such as age, sex, marital status, and education levels among parents were taken as covariates in the DID analysis to control for the possible confounding effects. A DID study is usually analyzed with a regression model, such as [21,22]:$$Y\;=\;intercept\;+\;\beta_{1}group\;+\;\beta_{2}time\;+\;\beta_{3}\left(group\;\times \;time\right)\;+\;\Sigma u\;+\;\varepsilon$$
where *Y* represents the dependent variable, that is, the KAP score. $\beta_{1}$ and $\beta_{2}$ are the coefficients of group and time, respectively. $\beta_{3}$ is the DID estimator, which represents the difference in the average value ($Y_{intervention}$) before and after implementing intervention in the intervention school, minus the difference in the average value ($Y_{control}$) before and after implementing intervention in the control school, that is, $\left(Y_{intervention,after\;}-\;Y_{intervention,before}\right)\;-\;\left(Y_{control,after}\;-\;Y_{control,before}\right)$. Σu represents the covariates including age and sex for students, and age, sex, education, and marital status for parents. *ε* is the random error term.

In order to compare the score changes among knowledge, attitude, and practice, we assigned a new indicator through a DID estimator divided by the full marks of each item, which was the standardized percentage of the increased scores.

The *t*-test and chi-square test were used to determine the homogeneity of study subjects between the intervention and control schools for the descriptive analysis, and to compare the KAP scores between different groups. Spearman correlation analysis was conducted to assess the correlations between KAP score changes after intervention and health education activities of the students at the intervention school, in order to identify the more effective intervention measures.

All the statistical analysis were performed using SAS9.4 (SAS Institute Inc., Cary, NC, USA).

## 3. Results

### 3.1. General Information of Subjects

The essential information of the primary school students was collected before and after intervention, as shown in Table 1. At the intervention school, 427 and 405 primary school students of third to fifth grades completed the questionnaire survey before and after the intervention, respectively; at the same time, 499 and 539 primary school students of third to fifth grades at the control school also completed the questionnaire survey during baseline and endline periods, respectively. The average age of the participated students at the intervention and control schools was 10.5 years and 10.4 years, respectively. The male students accounted for 52.9% and 53.1%, respectively, before and after intervention at the intervention school, whereas the percentage of male students at the control school was 59.0% and 58.4%, respectively. There was no significant difference in the age between the two schools at either the baseline or the endline. The percentage of the boys at the control school (59.0%) was significantly higher than that at the intervention school (52.9%) at the baseline, whereas no significant difference was found at the endline.

A total of 1264 parents were investigated at the two schools before and after intervention (Table 1). The age, sex ratio, and education level of the parents were significantly different between two schools at the baseline level.

### 3.2. Difference-in-Difference (DID) Analysis on KAP Score Changes of Primary School Students after Intervention

Table 2 shows the results of the DID analysis on KAP score changes (DID estimators) of students after intervention. Overall, the score changes for attitude and practice were smaller than that for knowledge. The scores for all the KAP items, except for the knowledge of climate change, were significantly increased; in particular, the score for knowledge increased 19.9% (95%CI: 16.3%, 23.6%) after intervention. The scores for attitude and practice increased 9.60% (95%CI: 5.35%, 13.9%) and 9.94% (95%CI: 8.26%, 18.3%), respectively. The total score for KAP increased 15.0% (95%CI: 12.1%, 17.8%) after intervention.

In terms of knowledge, the scores for cognition of heat waves, high temperature warning, and hot weather definition increased the most, by 44.7% (95%CI: 35.8%, 53.6%), 34.2% (95%CI: 24.7%, 43.7%), and 33.4% (95%CI: 27.4%, 39.4%), respectively. The scores for the awareness of climate change adaptation measures and heatstroke treatment measures increased 14.9% (95%CI: 8.94%, 20.9%) and 13.8% (95%CI: 8.78%, 18.9%). The relatively low change level for awareness of climate change and its health effects suggested that relevant education contents in the health education curriculum should be improved in future intervention campaigns.

For attitude and practice, the scores for willing to learn relevant knowledge and paying attention to weather forecasts increased relatively more, by 11.9% (95%CI: 6.79%, 16.9%) and 14.8% (95%CI: 8.48%, 21.1%), respectively.

Figure 1 shows the sex-specific KAP score changes (DID estimator) at the endline for the primary school students at the intervention school compared with the control school. It was found that all the KAP score changes of the male students were higher than those of the female students, even though there was no statistically significant change.

The age-specific KAP score changes (DID estimator) after intervention for the primary school students at the intervention school compared with the control school are shown in Figure 2. The results implied that older students had higher score changes, especially in knowledge and attitude. However, for appropriate practice, score changes increased with age until 11 years old.

### 3.3. Relationship between Health Education Activities and KAP Improvement of Primary School Students

In order to further evaluate the relationship between various health education activities and KAP improvement among primary school students, and to explore better intervention methods, Spearman correlation analysis was conducted between the changes in KAP scores and the participation in various health education activities at the intervention school. The results are shown in Table 3. All the health education activities at the intervention school in the program were significantly correlated with the changes in KAP scores of primary school students after the intervention. In particular, listening to lectures/watching cartoons, broadcasts between classes, hand-copying newspapers, and speech contests were more relevant. In contrast, attending none of the activities was negatively related to the score changes of all the KAP indicators.

Among the students that completed the questionnaire survey at the intervention school during the endline, 58.0% answered that they could obtain medicine to prevent heatstroke at the school clinic when they felt unwell during hot days. A share of 87.0% of the students believed that participating in these activities was helpful to increase their knowledge of climate change and health impacts, and 87.6% of the students indicated that they would be willing to participate in such activities again, indicating the impacts and effectiveness of the intervention activities.

The analysis of the differences in the age and sex ratio among students participating in different numbers of activities (Appendix A) demonstrated that older students participated in more activities than younger students (*p* < 0.0001). However, the sex difference was not statistically significant (*p* = 0.0783).

### 3.4. Intervention Effects on Parents

The KAP scores for parents of primary school students at the intervention and the control schools before and after intervention, and the results of the DID analysis of score changes, are shown in Table 4. The results of DID analysis show that all the score changes for knowledge, attitude, and practice were significantly increased after intervention; in particular, the score for knowledge increased by 22.5% (95%CI: 17.8%, 27.1%). Furthermore, the score changes for all of the knowledge items were positively significant; in particular, the scores for cognition of hot weather, high temperature warning, and heat wave increased the most, which was similar to the results of the students. The score for attitude and practice increased by 7.22% (95%CI: 0.96%, 13.5%) and 5.22% (95%CI: 0.73%, 9.71%), respectively. The total score for KAP increased by 14.8% (95%CI: 10.8%, 18.3%) after intervention.

Table 5 shows the estimates of effects of sex, age, marriage, and education on the KAP change of parents after intervention. It shows that the parents’ education level was positively related to the score change for knowledge, attitude, and practice, and that the score changes of females were higher than those of males.

## 4. Discussion

We assessed the effectiveness of implementing health education intervention program on climate change and health adaptation through a primary school-based intervention study in China using difference-in-difference (DID) analysis. The results indicated that not only the primary school students but also their parents had significantly improved the cognition of extreme heat and health adaptation.

To the best of our knowledge, this is the first primary school-based intervention program on improving KAP to adapt to extreme heat in China. Xu et al. performed a community-based intervention during heat waves to improve knowledge, attitude, and practice in Jinan, China [16]. Another community-based intervention program [20] was conducted to reduce the number of heat-related illnesses in Licheng, China. Both of these studies were focused on adults. Our findings provide robust evidence for intervention programs for children’s adaptation to heat waves and climate change.

We set up a standardized percentage of increased scores to facilitate comparison of the score changes among items. The itemized results of attitudes and practices of primary school students showed that changes in attitudes and behaviors were smaller than the improvement in knowledge. Smaller changes in attitude and practice scores may be because changes in attitude and behavior require more intensive and long-term education than the change in knowledge. Another possible reason may be associated with higher baseline scores due to the items being common questions or behaviors that most participants already performed [16], such as using air conditioning on hot days and washing hands before meals, which would make it difficult to distinguish differences after intervention. It was suggested that more questions should be designed in future studies to better evaluate the effect of the intervention on attitude and practice. Furthermore, the items with relatively lower changes suggested that the contents should be strengthened in future intervention, such as the contents regarding awareness of climate change and its health effects, in addition to long-term and reinforced health education, especially in enhancing transfer of knowledge change to attitude and behavior change. However, according to the standardized percentage of increased scores, the improvement in the use of air conditioning contributed to the second highest behavior score for both children and parents. It was shown that using air conditioning was an effective and feasible means of heatwave intervention.

We found greater changes, in particular, in knowledge and attitudes among older primary school students. For appropriate practice, however, score changes increased with age until 11 years old. This might may be because older students have higher capacity of knowledge acceptance, but find it difficult to easily change their behaviors once their own habits have been developed when reaching some certain age. It was further suggested that more efforts should be made to transfer knowledge to behavior change, and improving pupils’ behaviors or helping them develop good habits in future interventions. Schools have the best opportunity to develop curricula that prepare students for the anticipated changes to adapt to heat waves and climate change [23]. Our findings verified that good behavior should be developed at an early age, and health education to adapt to climate change should start with children. The findings also provided strong evidence for the inclusion of climate change and health in the curricula of primary schools.

It is notable in this study that we also evaluated the influence of implementing intervention in primary school students on the KAP of their parents. The results showed that the KAP, especially for knowledge items, of parents on heat waves and health were also significantly increased after implementing intervention for their children at the intervention school. It was found that the improvement in children’s cognition can promote parents’ awareness and cognition, and parents as the closest caregiver can, in turn, influence and improve children’s cognition, and urge children to develop good behavior, thus forming a virtuous circle. Strengthening health school programs is urgently needed to strengthen a country’s capability to successfully cope with the effects of climate change, according to a study on global health [24]. The successful practice of our study demonstrated that “small hands holding big hands” is an effective way to improve the awareness and behavior of children and their families to adapt to heat waves, and it has profound significance to enhance the national capacity to adapt to climate change. Moreover, it can also provide a successful experience for other public health education schemes, such as in the control and prevention of COVID-19.

All the health education activities in the program were significantly correlated with the changes in the KAP scores of primary school students after intervention. It was found that the health education activities had great impacts on the improvement and change in the relevant knowledge, attitude, and practice of primary school students. This was particularly the case of the health education curricula that included interesting activities, such as listening to lectures/watching cartoons, and experiential learning that can explore the activeness of students, such as broadcasts between classes, hand-copying newspapers, and speech contests. Curriculum and experiential learning approaches have also been proved to be effective strategies in nutrition health education of elementary school students [25]. Our results can provide a reference for the choice of intervention measures in the future when carrying out climate change adaptation-related intervention activities for primary school students.

We found the scores for most of the items of attitude and practice of students in the control school tended to decline after intervention compared with those before intervention. This may be related to the timing of the two surveys. The baseline survey was conducted in May when the weather was becoming warmer, whereas the endline survey was conducted in October when it was cooler. Different feelings about the weather may influence the attitude and practice for heat waves, which could induce information bias or memory bias. Even if these kinds of bias can be eliminated using the DID analysis, our results implied that the impacts of the time when questionnaire surveys on climate change-related perception estimation were performed should be considered, particularly for children who usually typically only remember short-term feelings.

Several limitations should be noted. First, the general information of participants in the intervention and controls school were not totally balanced and we only adjusted several observed factors, including age, sex, education, and marital status, in the DID analysis. Other possible factors should be further considered in future studies. Second, the pre- and post-intervention surveys were conducted during different seasons, which may induce information bias or memory bias. Third, in the study design, we did not group intervention measures according to the health education activities, so it is difficult to analyze the effects of different activities.

## 5. Conclusions

The effectiveness of a primary school-based intervention study in China was assessed using difference-in-difference (DID) analysis, with controls for age, sex, etc. The results demonstrated that not only the primary school students but also their parents’ cognition of extreme heat and health adaptation were significantly improved after health education and health promotion were implemented in the intervention school. The KAP improvement of boys and older students was higher than that of girls and younger students. For parents, the higher the education level, the higher the KAP improvement, and the increase was higher for females than for males. Health education in primary school, and particularly education that can influence students’ subjective activeness, was found to be an effective means to improve the cognition and behavior to adapt to heat waves and climate change for both primary school students and their parents. The results indicated that intervention to adapt to climate change should start from an early age in children, and particularly the measures for transferring knowledge to behavior change, and improving pupils’ behaviors or helping them develop good habits. Our findings provide strong evidence for the feasibility and importance of the inclusion of climate change and health in the primary school curricula. Moreover, our findings demonstrated that “small hands holding big hands” is an effective way to improve the awareness and behavior of children and their families to adapt to heat waves, and it has profound significance for enhancing the national capacity to adapt to climate change. The successful experience of our study can be generalized to respond to the increasing extreme weather/climate events in the context of climate change, such as heatwaves, and other emergent occasions or public health education, such as the control and prevention of COVID-19.

## Figures and Tables

**Figure 1 ijerph-19-02532-f001:**
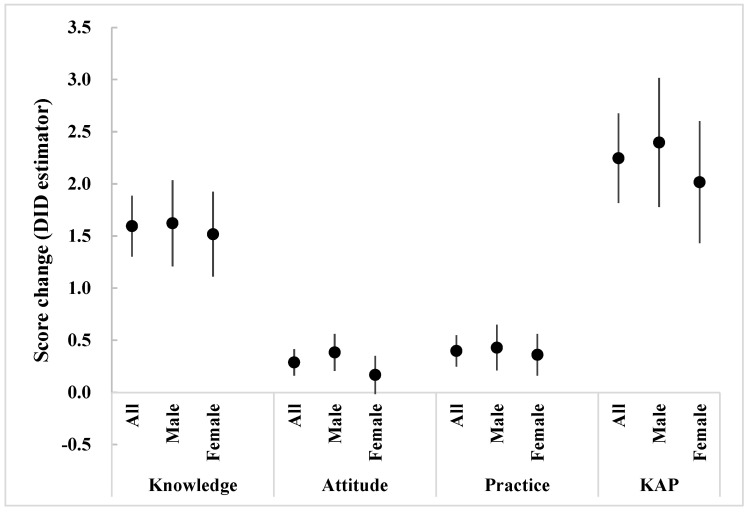
Sex-specific KAP score changes after intervention for students in the intervention school compared with the control school (adjusted with age by difference−in−difference (DID) analysis; KAP represents the total score of knowledge, attitude, and practice; score changes mean DID estimators).

**Figure 2 ijerph-19-02532-f002:**
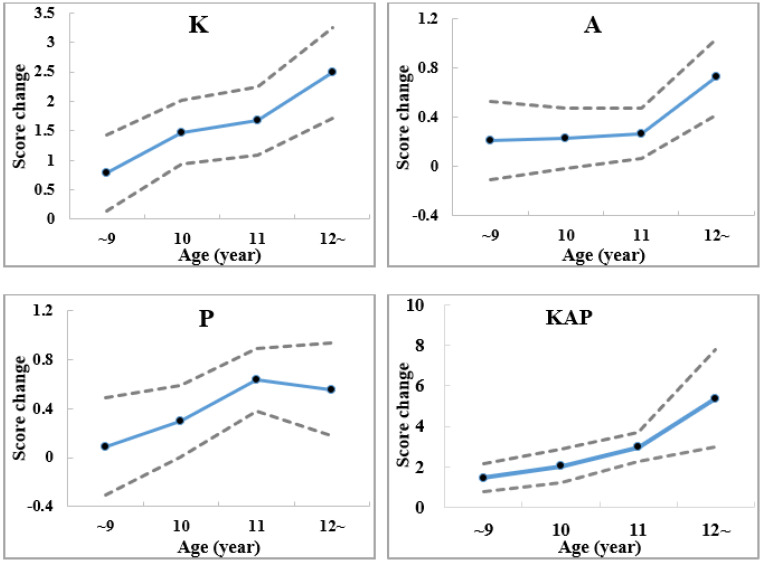
Age-specific KAP score changes after intervention for students in the intervention school compared with the control school (adjusted with sex by difference−in−difference (DID) analysis; the dotted lines represent the upper and lower limit of the 95% Confidence Interval of the score change; KAP represents the total score of knowledge, attitude, and practice; K: knowledge; A: attitude; P: practice; score changes mean DID estimators).

**Table 1 ijerph-19-02532-t001:** General information of the investigated primary students and parents in the intervention and control schools before and after intervention.

	Intervention School			Control School			^1^*p*-Value	^2^*p*-Value
	Before	After	*p*-Value	Before	After	*p*-Value		
Primary students								
N (Person)	427	405		499	539			
Age (Year)	10.5 ± 1.03	10.5 ± 1.00	0.3882	10.4 ± 1.03	10.4 ± 0.95	0.3515	0.1814	0.2484
Sex (Male, %)	52.9	53.1	0.9447	59	58.4	0.8678	0.0430 *	0.1123
Parents								
N (Person)	346	331		286	301			
Age (Year)	35.6 ± 6.82	36.6 ± 4.85	0.0259 *	37.6 ± 5.37	37.2 ± 5.90	0.4094	<0.0001 **	0.1804
Sex (Male, %)	42	48.6	0.0894	52.3	57.1	0.246	0.0128 *	0.0380 *
Nationality (Han Chinese, %)	99.4	99.7	1	99	99.7	0.3614	0.6627	1
Marriage								
Single	4	0	0.2242	1	3	0.6686	0.5955	0.1876
Married	329	320		273	284			
Devoiced	11	9		10	13			
Widowed	1	2		2	1			
Missing	1	-		-	-			
Education								
Primary schools and below	22	20	0.5468	35	25	0.0030 *	<0.0001 **	0.1059
Middle school	153	158		166	153			
Junior College and above	168	153		83	121			
Missing	3	-		-	2			

Note: ^1^
*p*-value represents the comparison between intervention and control schools before intervention, while ^2^
*p*-value represents the comparison between two schools after intervention. *: *p* < 0.05, **: *p* < 0.0001.

**Table 2 ijerph-19-02532-t002:** Scores for knowledge, attitude, and practice (KAP) for primary students in intervention and control schools before and after intervention and the difference-in-difference (DID) analysis of score changes.

KAP (Score)	Intervention School		Control School		Score change ^#^ (95%CI)	Standardized Percentage of Increased Scores ^$^ (95%CI, %)
	Before	After	Before	After		
KAP ^&^ (Full score: 15)	8.50 ± 2.02	10.8 ± 2.41	8.26 ± 2.64	8.54 ± 2.45	2.25 (1.82, 2.68) **	15.0 (12.1, 17.8)
Knowledge (K, Full score: 8)	3.30 ± 1.39	5.20 ± 1.52	3.10 ± 1.80	3.48 ± 1.61	1.59 (1.30, 1.89) **	19.9 (16.3, 23.6)
Climate change	0.84 ± 0.37	0.97 ± 0.18	0.65 ± 0.48	0.79 ± 0.40	0.03 (−0.04, 0.10)	2.71 (−4.44, 9.85)
Threshold of hot weather	0.04 ± 0.19	0.28 ± 0.45	0.13 ± 0.34	0.04 ± 0.18	0.33 (0.27, 0.39) **	33.4 (27.4, 39.4)
Heatwave	0.24 ± 0.43	0.70 ± 0.46	0.30 ± 0.46	0.31 ± 0.46	0.45 (0.36, 0.54) **	44.7 (35.8, 53.6)
The grading of heat warning	0.46 ± 0.50	0.73 ± 0.44	0.48 ± 0.50	0.44 ± 0.50	0.34 (0.25, 0.44) **	34.2 (24.7, 43.7)
Impacts of climate change	3.17 ± 2.35	4.58 ± 2.42	2.83 ± 2.47	3.60 ± 2.75	0.66 (0.17, 1.14) *	8.23 (2.17, 14.3)
Climate change mitigation measures	2.83 ± 1.80	4.25 ± 1.72	2.59 ± 1.83	3.14 ± 2.07	0.90 (0.54, 1.25) **	14.9 (8.94, 20.9)
Heatstroke symptom	3.82 ± 2.62	5.84 ± 2.47	3.43 ± 2.70	4.64 ± 3.04	0.81 (0.29, 1.33) *	8.99 (3.22, 14.8)
Heatstroke treatment	2.66 ± 1.56	3.67 ± 1.56	2.38 ± 1.69	2.54 ± 1.58	0.83 (0.53, 1.13) **	13.8 (8.78, 18.9)
Attitude (A, Full score: 3)	2.45 ± 0.61	2.62 ± 0.59	2.52 ± 0.60	2.43 ± 0.78	0.29 (0.16, 0.42) **	9.60 (5.35, 13.9)
Willing to learn relevant knowledge	7.77 ± 2.74	8.59 ± 2.28	8.04 ± 2.62	7.77 ± 2.81	1.19 (0.68, 1.69) **	11.9 (6.79, 16.9)
Willing to start from self	8.35 ± 2.34	8.78 ± 2.25	8.39 ± 2.44	7.99 ± 2.77	0.94 (0.45, 1.42) *	9.36 (4.54, 14.2)
Willing to change habits	8.40 ± 2.51	8.87 ± 2.25	8.74 ± 2.31	8.35 ± 2.67	0.99 (0.52, 1.47) **	9.94 (5.22, 14.7)
Practice (P, Full score: 4)	2.76 ± 0.73	3.06 ± 0.72	2.69 ± 0.89	2.63 ± 0.84	0.40 (0.25, 0.55) **	9.94 (8.26, 18.3)
Pay attention to weather forecast	0.59 ± 0.31	0.71 ± 0.30	0.59 ± 0.34	0.57 ± 0.34	0.15 (0.08, 0.21) **	14.8 (8.48, 21.1)
Appropriate behavior in hot weather	1.45 ± 0.82	1.72 ± 0.71	1.45 ± 0.91	1.45 ± 0.79	0.23 (0.07, 0.38) *	7.51 (2.42, 12.6)
Use air conditioning	0.89 ± 0.31	0.93 ± 0.26	0.86 ± 0.35	0.83 ± 0.38	0.09 (0.02, 0.15) *	8.87 (2.50, 15.2)
Wash hands before meals	0.80 ± 0.40	0.85 ± 0.35	0.77 ± 0.42	0.75 ± 0.43	0.09 (0.01, 0.17) *	8.83 (1.10, 16.6)

Note: KAP ^&^ represents the total score of knowledge, attitude, and practice; **^#^**: estimator of DID analysis. **^$^**: score change divided by full score; *****: *p* < 0.05; ******: *p* < 0.0001.

**Table 3 ijerph-19-02532-t003:** The relationship between KAP score changes after intervention and health education activities of pupils in the intervention school.

Health Education Activities	KAP ^&^		Knowledge		Attitude		Practice	
	r	*p*-Value	r	*p*-Value	r	*p*-Value	r	*p*-Value
Listen to lectures or watch cartoons	0.307	<0.0001	0.245	<0.0001	0.183	0.0004	0.112	0.0313
Recess radio	0.263	<0.0001	0.195	0.0002	0.176	0.0007	0.151	0.0035
Speech contest	0.258	<0.0001	0.203	<0.0001	0.124	0.0170	0.147	0.0047
Hand-copying newspapers	0.245	<0.0001	0.202	0.0001	0.068	0.1958	0.173	0.0009
Make blackboard newspapers	0.233	<0.0001	0.164	0.0016	0.089	0.0888	0.180	0.0005
Theme essay contest	0.226	<0.0001	0.154	0.0031	0.146	0.0049	0.128	0.0136
Drawing contest	0.209	<0.0001	0.107	0.0399	0.132	0.0113	0.149	0.004
Social activities during summer vacation	0.209	<0.0001	0.138	0.0083	0.166	0.0014	0.100	0.0535
Perform a skit	0.161	0.0021	0.100	0.0569	0.085	0.1024	0.141	0.0067
Attended none of the above	−0.211	<0.0001	−0.144	0.0059	−0.113	0.0297	−0.108	0.0385

Note: KAP ^&^ represents the total score of knowledge, attitude, and practice; r is the correlation coefficient.

**Table 4 ijerph-19-02532-t004:** Scores of knowledge, attitude, and practice (KAP) for parents of primary students at intervention and control schools before and after intervention and the difference-in-difference (DID) analysis of score changes.

KAP	Intervention School		Control School		Score Change ^#^ (95%CI)		Standardized Percentage of Increased Scores ^$^ (95%CI, %)	
	Before	After	Before	After				
KAP ^&^ (Full score: 15)	9.18 ± 2.64	11.5 ± 2.74	9.47 ± 2.61	9.61 ± 2.85	2.22	(1.62, 2.82) **	14.8	(10.8, 18.3)
Knowledge (Full score: 8)	4.19 ± 1.61	5.88 ± 1.76	4.49 ± 1.59	4.43 ± 1.72	1.80	(1.43, 2.17) **	22.5	(17.8, 27.1)
Climate change	0.90 ± 0.30	0.94 ± 0.23	0.91 ± 0.28	0.87 ± 0.33	0.09	(0.03, 0.16) *	9.22	(2.79, 15.6)
Heat warning	0.62 ± 0.48	0.84 ± 0.36	0.68 ± 0.47	0.62 ± 0.48	0.28	(0.18, 0.38) **	27.9	(17.8, 37.9)
Threshold of hot weather	0.24 ± 0.43	0.51 ± 0.50	0.31 ± 0.46	0.20 ± 0.39	0.40	(0.29, 0.50) **	39.5	(29.4, 49.6)
Heatwave	0.36 ± 0.48	0.71 ± 0.45	0.39 ± 0.49	0.47 ± 0.49	0.28	(0.17, 0.38) **	27.6	(16.9, 38.2)
Impacts of climate change	3.59 ± 2.40	5.41 ± 2.62	3.78 ± 2.68	4.00 ± 2.64	1.66	(1.09, 2.24) **	20.8	(13.6, 28.0)
Climate change mitigation measures	3.14 ± 1.83	4.15 ± 1.92	3.50 ± 1.91	3.32 ± 2.01	1.25	(0.82, 1.67) **	20.8	(13.8, 27.9)
Heatstroke symptom	4.38 ± 2.42	6.54 ± 2.65	4.65 ± 2.70	5.06 ± 2.67	1.81	(1.23, 2.39) **	20.1	(13.7, 26.6)
Heatstroke treatment	3.07 ± 1.47	3.89 ± 1.43	3.13 ± 1.55	3.27 ± 1.55	0.68	(0.34, 1.01) **	13.5	(6.88, 20.2)
Attitude (Full score: 3)	2.38 ± 0.87	2.57 ± 0.78	2.46 ± 0.87	2.45 ± 0.86	0.22	(0.03, 0.40) *	7.22	(0.96, 13.5)
Willing to pay attention to climate change	0.87 ± 0.34	0.93 ± 0.26	0.88 ± 0.32	0.90 ± 0.30	0.05	(−0.02, 0.12)	4.87	(−2.00, 11.7)
Be interested in relevant knowledge	0.81 ± 0.39	0.85 ± 0.36	0.80 ± 0.40	0.81 ± 0.39	0.03	(−0.05, 0.12)	3.16	(−5.42, 11.7)
Willing to attend relevant activities	0.70 ± 0.46	0.79 ± 0.41	0.78 ± 0.42	0.74 ± 0.44	0.14	(0.04, 0.23) *	13.6	(4.09, 23.2)
Practice (Full score: 4)	2.61 ± 0.81	3.03 ± 0.80	2.51 ± 0.78	2.72 ± 0.85	0.21	(0.03, 0.39) *	5.22	(073, 9.71)
Pay attention to weather forecast	0.91 ± 0.29	0.94 ± 0.24	0.90 ± 0.30	0.93 ± 0.26	0.00	(−0.06, 0.06)	−0.40	(−6.44, 5.64)
Use air conditioning	0.86 ± 0.35	0.95 ± 0.21	0.83 ± 0.38	0.87 ± 0.34	0.06	(−0.01, 0.13)	6.02	(−1.16, 13.2)
Actively consult impacts of weather	0.41 ± 0.49	0.52 ± 0.50	0.42 ± 0.49	0.48 ± 0.50	0.05	(−0.06, 0.16)	5.05	(−6.03, 16.1)
Appropriate activities in hot weather	3.05 ± 2.42	4.32 ± 2.44	2.56 ± 2.54	3.14 ± 2.34	0.69	(0.15, 1.23) *	9.90	(2.20, 17.6)

Note: KAP ^&^ represents the total score of knowledge, attitude, and practice; **^#^**: estimator of DID analysis; **^$^**: score change divided by full score; *: *p* < 0.05; **: *p* < 0.0001.

**Table 5 ijerph-19-02532-t005:** The estimates of effects of sex, age, marriage, and education on KAP change of parents after intervention, from DID analysis.

Outcomes	Variables	Estimates	SD ^$^	t Value	*p* Value
KAP ^&^					
	Sex	0.32	0.16	2.05	0.0405
	Age	0.00	0.01	−0.15	0.8808
	Marriage	−0.05	0.32	−0.17	0.8642
	Education	0.40	0.09	4.26	<0.0001
	DID ^#^	2.22	0.31	7.26	<0.0001
Knowledge					
	Sex	0.07	0.10	0.75	0.4521
	Age	0.00	0.01	−0.46	0.6474
	Marriage	−0.11	0.20	−0.55	0.5858
	Education	0.25	0.06	4.3	<0.0001
	DID ^#^	1.80	0.19	9.49	<0.0001
Attitude					
	Sex	0.12	0.05	2.51	0.0121
	Age	0.00	0.00	0.56	0.5774
	Marriage	0.06	0.10	0.65	0.5142
	Education	0.07	0.03	2.4	0.0167
	DID ^#^	0.22	0.10	2.26	0.0240
Practice					
	Sex	0.12	0.05	2.67	0.0076
	Age	0.00	0.00	−0.14	0.8891
	Marriage	−0.01	0.09	−0.13	0.8979
	Education	0.08	0.03	2.85	0.0045
	DID ^#^	0.21	0.09	2.28	0.0229

Note: KAP ^&^ represents the total score of knowledge, attitude, and practice; **^#^**: estimator of DID analysis; ^$^: standard deviation.

## Data Availability

The datasets generated and/or analysed during the current study are not publicly available due to including personal information but are available from the corresponding author on reasonable request.

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
