# Peer review of "Effectiveness Evaluation of a Primary School-Based Intervention against Heatwaves in China"

_ijerph, 2022, doi:10.3390/ijerph19052532_

Round 1

Reviewer 1 Report

The authors have responded to my comments. Two comments are provided as follows.

1.The content in the conclusion still did not present important results. In section of 3. Results, four items were discussed. Authors are encouraged to provide a brief description of these research findings so that readers can succinctly understand what the research findings are about.

2.The content of the abstract should explain the motivation, methods and brief results of the research. At the same time, it can correspond to the content of the conclusion. Therefore, the content of the abstract needs to be re-examined and adjusted.

Author Response

1.The content in the conclusion still did not present important results. In section of 3. Results, four items were discussed. Authors are encouraged to provide a brief description of these research findings so that readers can succinctly understand what the research findings are about.

Response: Thank you for the comments. We have added a brief description of our findings in the conclusion, as

"The effectiveness of a primary school-based intervention study in China was assessed by using difference-in-difference (DID) analysis with controlling for age, sex, etc. The results demonstrated that not only the primary school students but also their parents’ cognition of extreme heat and health adaptation were significantly improved after health education and health promotion were implemented in the intervention school. The KAP improvement of boys and older students were higher than that of girls and younger students. For parents, the higher the education level, the higher KAP improvement, and the female was improved greater than the male. Health education in primary school, especially those can exert students’ subjective activeness, was an effective way to improve the cognition and behavior to adapt to heat waves and climate change for both primary school students and their parents. ".

2.The content of the abstract should explain the motivation, methods and brief results of the research. At the same time, it can correspond to the content of the conclusion. Therefore, the content of the abstract needs to be re-examined and adjusted.

Response: Thank you for the comments. We have revised the abstract as suggested.

Reviewer 2 Report

Thank you for your great job. For your information. Let me tell you a meaning of ICC.

An ICC is abbreviation of intraclass correlation coefficient. When you plan to conduct a cluster randomised controlled trial, the ICC should be considered to calculate a priori sample size.

Author Response

Thank you for your great job. For your information. Let me tell you a meaning of ICC.

An ICC is abbreviation of intraclass correlation coefficient. When you plan to conduct a cluster randomised controlled trial, the ICC should be considered to calculate a priori sample size.

Response: Thank you so much for your kind instruction and explanation. It is very helpful for our similar studies in the future.

This manuscript is a resubmission of an earlier submission. The following is a list of the peer review reports and author responses from that submission.

Round 1

Reviewer 1 Report

In this paper, the authors randomly selected two primary schools in Dongtai of Jiangsu Province in China to conduct health education surveys at the intervention school concerning the changes in students’ awareness and capability on responding to extreme heat from May to September in 2017. It is found that the score of knowledge, attitude, and practice of students have improved. The impact of extreme high temperature on national health has become one of the important topics actively discussed in many countries. However, some points need to be clarified or explained before being considered for publication. Below are further elaborations of my viewpoints:

  1. In the section of 1. Background, the authors mentioned that they hope the findings could have policy implications to other countries and regions with similar climatic and socioeconomic situations. However, what kind of socioeconomic situation was not mentioned. This point needs further explanation. It is better to have quantitative data as a reference.
  2. Moreover, what the main economic activities and characteristics of the studied city are need to be described. For example, it is an agricultural city, a heavy industry manufacturing city, or a tourist city, etc.
  3. In the section of 3.4. Intervention Effects on Parents, the authors are encouraged to further explain which measures directly affect children, and which ones affect the parents first and then affect the children throughout the process.
  4. Please describe what the the effect of the parents’ education level on the results.
  5. In the section of 4. Discussion, please indicate the degree of influence of "whether the air conditioning system is used" on the results of this study.
  6. The description of the conclusion is somewhat brief and does not contain important results of this paper. Please reorganize the content again.

Author Response

In this paper, the authors randomly selected two primary schools in Dongtai of Jiangsu Province in China to conduct health education surveys at the intervention school concerning the changes in students’ awareness and capability on responding to extreme heat from May to September in 2017. It is found that the score of knowledge, attitude, and practice of students have improved. The impact of extreme high temperature on national health has become one of the important topics actively discussed in many countries. However, some points need to be clarified or explained before being considered for publication. Below are further elaborations of my viewpoints:

1. In the section of 1. Background, the authors mentioned that they hope the findings could have policy implications to other countries and regions with similar climatic and socioeconomic situations. However, what kind of socioeconomic situation was not mentioned. This point needs further explanation. It is better to have quantitative data as a reference. Moreover, what the main economic activities and characteristics of the studied city are need to be described. For example, it is an agricultural city, a heavy industry manufacturing city, or a tourist city, etc.

Response: Thank you very much for this great advice. Since socioeconomic issues are a very complex question and not addressed in our manuscript, we have decided to delete the words “socioeconomic situation” and revised as “We hope the findings could have policy implications to other countries and regions with similar climatic situations.”

2. In the section of 3.4. Intervention Effects on Parents, the authors are encouraged to further explain which measures directly affect children, and which ones affect the parents first and then affect the children throughout the process.

Response: Thank you for this suggestion. Actually, all the intervention measures were conducted to the students in school and affected the students first, and then affected the parents through their children.

3. Please describe what the effect of the parents’ education level on the results.

Response: Thank you for this suggestion. We have listed the estimates of effects of sex, age, marriage and education on KAP change of parents after intervention in Table 5, and added this description at the second paragraph in section of 3.4 of the results part as “Table 5 shows the estimates of effects of sex, age, marriage and education on KAP change of parents after intervention. It was showed that the parents’ education level was positively related to the score change of knowledge, attitude and practice. And the score changes of females were higher than those of males.

4. In the section of 4. Discussion, please indicate the degree of influence of "whether the air conditioning system is used" on the results of this study.

Response: Thank you for this kind suggestion. We have added the contents at the third paragraph in Discussion part, as “However, according to the standardized percentage of increased scores, the improvement of using air conditioning contributed to the second in behavior score for both children and parents. It was implied that using air conditioning was an effective and feasible way of heatwave intervention.”

5. The description of the conclusion is somewhat brief and does not contain important results of this paper. Please reorganize the content again.

Response: Thank you for this kind suggestion. We have added some contents in the conclusion, such as “Intervention to adapt to climate change should start from the early stage of children, especially the measures for transferring knowledge to behavior change and improving pupils’ behaviors or helping them develop good habits.”, “our findings demonstrated that "small hands holding big hands" is an effective way to improve the awareness and behavior of children and their families to adapt to heat waves, and it has profound significance to enhance the national capacity to adapt to climate change”.

Reviewer 2 Report

Dear Authors,

This study is to evaluate a primary school based intervention against heatwaves in China. The contents is interesting, but I will point to the issues to be discussed.

・Line 91 proximately should be changed to approximately.

・Methods : Authors randomly chose two primary schools and allocated one school in the intervention group and the other school in the control group. Author should state how to determine number of schools based on a priori sample size calculation. If this study is a cluster randomised controlled trial, an ICC should be also considered.

・When authors report the result of randomized controlled trial, they should follow the CONSORT statement. Please check the checklist.

・According to Figure A1, participants of the intervention group did not necessarily participated in all activities, meaning that program attended varied in each participants. Authors should mention whether this difference affect the results, and whether authors take measures in order to encourage participants to attend every activity.

・In addition to the abovementioned concern, authors should state how to select the participants who followed the intervention program. In addition, authors should mention that intention-to-treat analysis was performed in this study, if appropriate. In addition, whether difference in demographic characteristics were different between participants who joined more activities and those who did not.

・Authors mentioned that participants in the control group received no program, which might cause inequality between groups. Did authors make any ethical consideration for participants who were allocated to the control group?

・Results 3.3 Authors showed relationship between each health education activity and KAP improvement. However, some participants who attended one activities did participated in other certain program and others did not. Therefore, I think that effect of each domain of the intervention program is difficult in this study design. It is better to describe this point in the limitation part.

・Explanation regarding DID analysis starting from Line 310 is detailed and understandable, however seems somewhat redundant. Authors can delete the part.

・Authors stressed that this kind of intervention should be implemented in the education field for primary school children. When we consider for future perspective, how do the authors think about who should responsible for implementation and when and how the program should be done. 

・There are several typos (for example : lack of %) are found. Please check again.

Author Response

This study is to evaluate a primary school based intervention against heatwaves in China. The contents is interesting, but I will point to the issues to be discussed.

・Line 91 proximately should be changed to approximately.

Response: Thank you. The “proximately” has been be changed to “approximately”.

・Methods : Authors randomly chose two primary schools and allocated one school in the intervention group and the other school in the control group. Author should state how to determine number of schools based on a priori sample size calculation. If this study is a cluster randomised controlled trial, an ICC should be also considered.

Response: Thank you for the comments. We selected two primary schools located in the two communities with similar economic and education levels in Dongtai on the basis of their willingness of participation. One school was randomly chosen as the intervention school and the other one as the control school. As the lacking of related studies, we did not calculate the sample size by traditional formula. The sample size was decided according to several experts, who suggested 400 students for intervention group and 400 students for control group. I am sorry, I do not know what ICC mean, and would you please tell us the complete spelling? Thank you so much.

・When authors report the result of randomized controlled trial, they should follow the CONSORT statement. Please check the checklist.

Response: Thank you for the comments. We have added the CONSORT checklist.

・According to Figure A1, participants of the intervention group did not necessarily participated in all activities, meaning that program attended varied in each participants. Authors should mention whether this difference affect the results, and whether authors take measures in order to encourage participants to attend every activity.

Response: Actually, the education points were very similar or almost same among different activities. So, even participants of the intervention group did not participated in all activities, they obtained the similar education effects, which can be verified from the results in Table 3, which shows the very similar correlation coefficients of relationships between KAP score changes after intervention and different health education activities of pupils in intervention school. However, it is really a limitation in the study design that we did not grouping intervention measures according to the health education activities, so that it is difficult to analyze the effects of different kinds of activities. We have added this point in the limitation part.

・In addition to the abovementioned concern, authors should state how to select the participants who followed the intervention program. In addition, authors should mention that intention-to-treat analysis was performed in this study, if appropriate. In addition, whether difference in demographic characteristics were different between participants who joined more activities and those who did not.

Response: On the basis of their willingness to participate, we selected two primary schools located in the two communities with similar economic and education levels in Dongtai. One school was randomly chosen as the intervention school and the other one as the control school. Considering their understanding ability, all the students of the third to the fifth grades at each school were chosen as the study population. As this study was based on the questionnaire analysis, so all the questionnaires that we collected from the students and their parents in intervention school were taken into the intervention group, regardless of whether they had participated in the health education activities or not. Same did for the control group.

We have analyzed the difference in age and sex ratio among students participated in different number of activities and the results were shown in Table A3, and added the contents at the section 3.3 of results part, as follows:

The analysis on difference in age and sex ratio among students participated in different numbers of activities (Table A3) demonstrated that older students participated in more activities than younger students (P<0.0001). However, the sex difference was not statistically significant (P=0.0783)”.

・Authors mentioned that participants in the control group received no program, which might cause inequality between groups. Did authors make any ethical consideration for participants who were allocated to the control group?

Response: Thank you for this comments. We have provide the same intervention to control school after this intervention program.

・Results 3.3 Authors showed relationship between each health education activity and KAP improvement. However, some participants who attended one activities did participated in other certain program and others did not. Therefore, I think that effect of each domain of the intervention program is difficult in this study design. It is better to describe this point in the limitation part.

Response: We have added this point in the limitation part as “Third, in the study design, we did not grouping intervention measures according to the health education activities, so it is difficult to analyze the effects of different activities.”.

・Explanation regarding DID analysis starting from Line 310 is detailed and understandable, however seems somewhat redundant. Authors can delete the part.

Response: We have deleted this part as suggested.

・Authors stressed that this kind of intervention should be implemented in the education field for primary school children. When we consider for future perspective, how do the authors think about who should responsible for implementation and when and how the program should be done. 

Response: From the results, we concluded that intervention on climate change adaptation should be implemented in the education field for primary school children. And we hope our findings could provide reference and evidence for government or decision-makers, especially the education departments, to make education-related policies, such as inclusion of climate change and health in the primary school curricula. And we think it is a job that should be implemented by the cooperation of multiple departments, including education department, health department, environmental department, etc. In the context of global climate change, the sooner this program is conducted, the better.

・There are several typos (for example : lack of %) are found. Please check again.

Response: Thank you very much and sorry for this mistake. We have checked and added % in the manuscript.